# Evaluation of Drifting Snow Susceptibility Based on GIS and GA-BP Algorithms

**Bohu He [1], Mingzhou Bai [1,2,*], Binglong Liu [3], Pengxiang Li [1], Shumao Qiu [1], Xin Li [1] and Lusheng Ding [4]**

1   School of Civil Engineering, Beijing Jiaotong University, Beijing 100044, China; 18115063@bjtu.edu.cn (B.H.); 19115058@bjtu.edu.cn (P.L.); 13115320@bjtu.edu.cn (S.Q.); 16115268@bjtu.edu.cn (X.L.)
2   Key Laboratory of Track Engineering, Beijing Jiaotong University, Beijing 100044, China
3   Qingdao Municipal Engineering Design and Research Institute, Qingdao 266000, China; 18125908@bjtu.edu.cn
4   Geological Subgrade Design Branch, Xinjiang Railway Survey and Design Institute, Urumqi 830011, China; 20125971@bjtu.edu.cn
*   Correspondence: mzhbai@bjtu.edu.cn

**Abstract:** Drifting snow, the flow of dispersed snow particles near ground level under the action of wind, is a major form of snow damage. When drifting snow occurs on railways, highways, and other transportation lines, it seriously affects their operational safety and results in drifting snow disasters. Drifting snow disasters frequently occur in the high latitudes of northwest China. At present, most scholars are committed to studying the prevention and control measures of drifting snow, but the prerequisite for prevention is to effectively evaluate the susceptibility of drifting snow along railways and highways to identify areas with a high risk of occurrence. Taking the Xinjiang Afukuzhun Railway as an example, this study uses a geographic information system (GIS) combined with on-site monitoring and surveys to establish a drifting snow susceptibility evaluation index system. The drifting snow susceptibility index (DSSI) is calculated through the weight of an evidence (WOE) model, and a genetic algorithm backpropagation (GA-BP) algorithm is used to obtain optimised evaluation index weights to improve the accuracy of model evaluation. The results show that the accuracies of the WOE model, WOE backpropagation (WOE-BP) model, and weight of evidence genetic algorithm backpropagation (WOE-GA-BP) model are 0.747, 0.748, and 0.785, respectively, indicating that the method can be effectively applied to evaluate drifting snow susceptibility.

**Keywords:** GIS; drifting snow; GA-BP; WOE; susceptibility

## 1. Introduction

Drifting snow is an atypical gas–solid two-phase flow in which dispersed snow particles move near the ground under the action of wind; also referred to as snowstorm flow [1,2], it is a major form of snow damage. In the first instance, drifting snow can be characterised according to the blowing height: (i) low drifting snow, where the blowing height is within the range of 0–2 metres near the surface; (ii) blowing snow, where the height of blowing is more than 2 metres; (iii) snowstorm, which occurs in extremely low temperature and extreme wind speed conditions, and during which visibility is extremely low [3]. An alternative characterisation is based on the motion of snow particles in drifting snow, which can be divided into three groups based on different moving states: creep, jump, and suspension [4]. Drifting snow is mainly distributed in China, Russia, the United States, and other countries with a large latitude span. In China, it is distributed in Xinjiang, Northeast China, Tibet, and other regions [5]. Drifting snow severely affects transportation, agricultural and animal husbandry production, industrial construction, and people's life and property safety [6–12].

The snowstorms of February 1978 in England resulted in $1 billion in damage. In October 1985, a drifting snow disaster affected the Qinghai–Tibet Plateau, and the affected area was equivalent to the combined area of China's Jiangsu and Shandong provinces. In

February 2009, the UK was hit by the worst snowstorm in the century, with more than 3000 schools closed along and major closures in the road, rail, tube, and air transport systems, costing the economy nearly 1.2 billion pounds [13]. On 3 January 2010, a sudden snowstorm in Inner Mongolia led to heavy snow accumulation on the railway line; more than 1817 trains were stopped, and several carriages were buried in snow, trapping more than 1400 passengers [14]. On 22 December 2012, under the double action of extreme cold and strong wind, a section of the G30 Connection Line in Xinjiang was hit by drifting snow. More than 300 vehicles were blocked, and hundreds of people were trapped on a six-kilometre section. In February 2013, a major snowstorm, Nemo, hit the northeastern part of the United States, which caused five states to declare a state of emergency; all highways in Massachusetts were closed [15]. From February 29 to 13, 2016, severe drifting snow disasters frequently occurred in Maytas, Xinjiang Region, causing traffic jams and trapping hundreds of cars and passengers in disaster areas. With the acceleration of construction in northwestern China, road construction has presented an urgent need for the risk assessment and effective protection of drifting snow disasters.

From different dynamic factors, snow disasters can be classified into two categories. One is the disaster resulting from the accumulation of natural snow to a certain thickness, and the other is the drifting snow disaster in certain areas resulting from the transport of wind. Several scholars have studied the obstructive effect of objects in drifting snow [16,17], and the relationship between wind speed, snow particle size, and snow output [18–21]. Early snow disaster assessment research was based on a combination of satellite data and fuzzy mathematics method theory [22–28]. With the continuous development of geographic information systems (GISs), several scholars have begun to use them to assess the risk of snow disasters. For example, snow disaster factors have been extracted and combined with a grey weighting clustering method and an analytic hierarchy process to establish a snow disaster assessment model and snow disaster risk-zoning analysis [29–31].

However, the existing research results have notable shortcomings for drifting snow, especially for railway drifting snow disasters. Firstly, the influencing factors of linear engineering drifting snow are notably different from natural snow disasters, but recent studies do not specifically establish a risk evaluation index system for railway drifting snow disasters [32–34]. In addition, risk research is focused on snow and the resultant situation of road icing [35–39]. Secondly, while establishing the index evaluation system, there is no objective analysis of the weight of each index; moreover, probability and fuzzy analyses are the main methods used for developing the system [40,41].

This study uses the Xinjiang Afuzhun Railway as the research object, combined with a site survey and monitoring data based on a GIS platform, establishes a railway drifting snow susceptibility evaluation system, and uses the weight of evidence (WOE) model to calculate the susceptibility index. Then combined with a genetic algorithm backpropagation (GA-BP) algorithm, the weight of each index factor was studied to optimise the evaluation results, and the reliability of the method was verified by a receiver operating characteristic (ROC) curve. The result shows that this method can provide a reference for other similar railway projects.

## 2. Evaluation Model

### 2.1. Evidence Weight Method

The WOE method is based on Bayesian statistical theory. The method uses statistical analysis of the contribution of the evidence-level factors to the research goal to predict whether the event will occur; in this manner, the influence of subjective factors can effectively be avoided [42]. This method was first applied in the field of medicine and then introduced by geologists Bonham-Carter et al. [43] and Ahterberg et al. [44] into the field of mineral research. It has been widely used in research on landslides, debris flows and other geological hazards [45]; however, it has seldom been applied to drifting snow disaster evaluation.

The appropriate starting place to describe the mathematical principle of WOE is to consider the study area as being equally divided into N grids, where $M$ represents the total number of grids with drifting snow, and $\overline{M}$ represents the total number of grids with no drifting snow. The number of grids with drifting snow in the secondary state classification of a certain evidence layer is represented by $A$, and event $A$ represents the number of drifting snow grids that did not occur. For the secondary status of any evidence factor, its contribution to the drifting snow is defined as

$$W_i^+ = ln\left[P\left(\frac{A}{M}\right)/P\left(\frac{A}{\overline{M}}\right)\right] \tag{1}$$

$$W_i^- = ln\left[P\left(\frac{\overline{A}}{M}\right)/P\left(\frac{\overline{A}}{\overline{M}}\right)\right] \tag{2}$$

where *P(A/M)* is the conditional probability, which is the probability of $A$ occurring under $M$.

The computed values of $W_i^+$ and $W_i^-$ can indicate the impact of the secondary state classification in the evidence layer on the occurrence of drifting snow. Specifically, $W_i^+ > 0$ or $W_i^- < 0$ indicates that the grading factor is conducive to the occurrence of drifting snow, whereas $W_i^+ < 0$ or $W_i^- > 0$ alternatively indicates that it is not conducive to the occurrence of drifting snow. The difference between the two can indicate the strength of the correlation between the secondary factor and drifting snow, that is, $W_{fi} = W_i^+ - W_i^-$. The larger $W_{fi}$ is, the better the indicator of this secondary factor is to the occurrence of drifting snow. Conversely, if the indicator is poor, then the secondary factor is not conducive to the occurrence of drifting snow. If $W_{fi} = 0$, the secondary factor has no effect on drifting snow.

### 2.2. Coupling Model

The relationship between the various factors influencing wind and snow and the occurrence of drifting snow is complex and nonlinear; it cannot be accurately studied using functional relationships. However, BP neural network has good applicability to nonlinear problems and can be used for drifting snow disaster evaluation [46]. In previous studies, it has been mostly used for landslide risk evaluation. The factors that affect the stability of landslides are used as an input layer, and the risk index is used as an output layer. However, the consideration for the connection between neurons was slightly lacking [47,48]. The genetic algorithm can optimise the weights and thresholds of the neural network while making full use of the nonlinear mapping ability of the neural network and improving the convergence speed and prediction accuracy of the neural network [49]. Therefore, the weight of each input layer in the improved BP neural network can be coupled with the evidence weight model to obtain the drifting snow susceptibility index (DSSI), and the susceptibility zone map can be obtained using the GIS platform. The calculation process is shown in Figure 1.

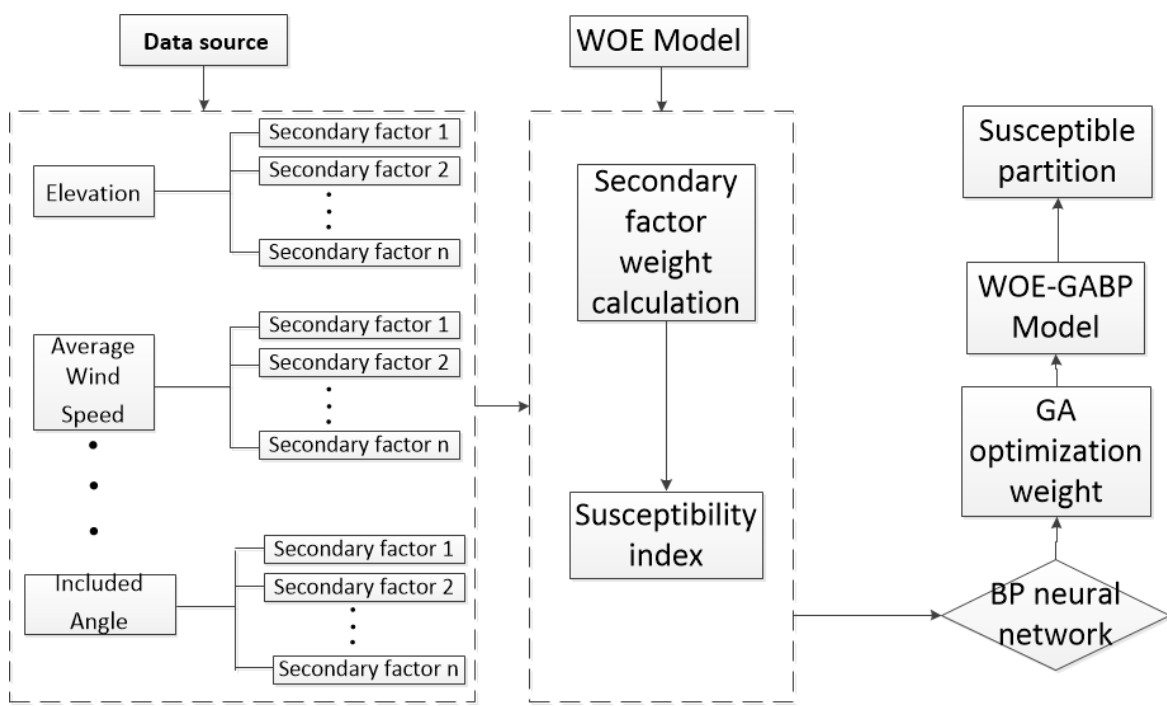

**Figure 1.** Calculation flow chart.

## 3. Overview of the Study Area and Data Sources

### 3.1. Overview of the Study Area

The A(Aletai)-F(Fuyun)-Z(Zhun-dong) Railway is located in Changji Autonomous Prefecture of the Aletai Region, Xinjiang Uygur Autonomous Region. It runs across one city, two countries, and eight administrative regions. The line lies between 45°11–48°11′ N and 87°38–90°32′ E with an overall altitude of 597–1219 m, where the low terrain is generally to the north, and the high terrain is overall to the south. From northwest to southeast, it passes through two geomorphic units, that is, the intermontane alluvial plain and the low-hill area at the southern foot of the Altai Mountains (Figure 2).

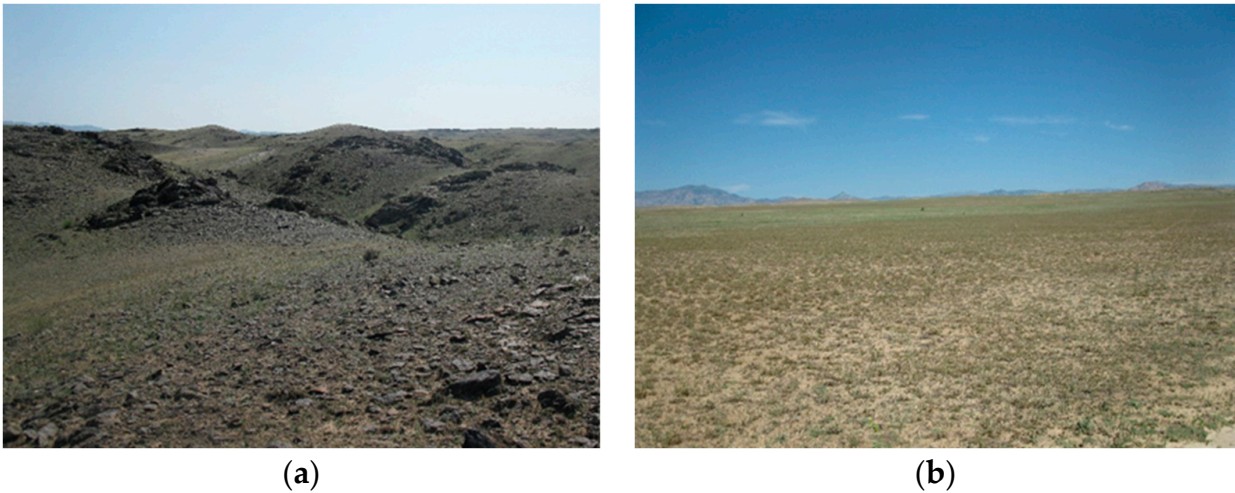

(**a**)                      (**b**)

**Figure 2.** Geomorphological unit. (**a**) Hilly landform, (**b**) Hilly plain.

The AFZ Railway (Figure 3) passes through the zone between the northwest area of the Zhungeer Basin and the southern foot of the Altai Mountains, which belongs to the temperate continental climate. There is little rain in summer, and the winters are cold and long. The north–south span of the whole line is large and located in high latitude

areas. The area has deep snow in winter with high-wind speed; it is one of the three major snow-covered areas in China.

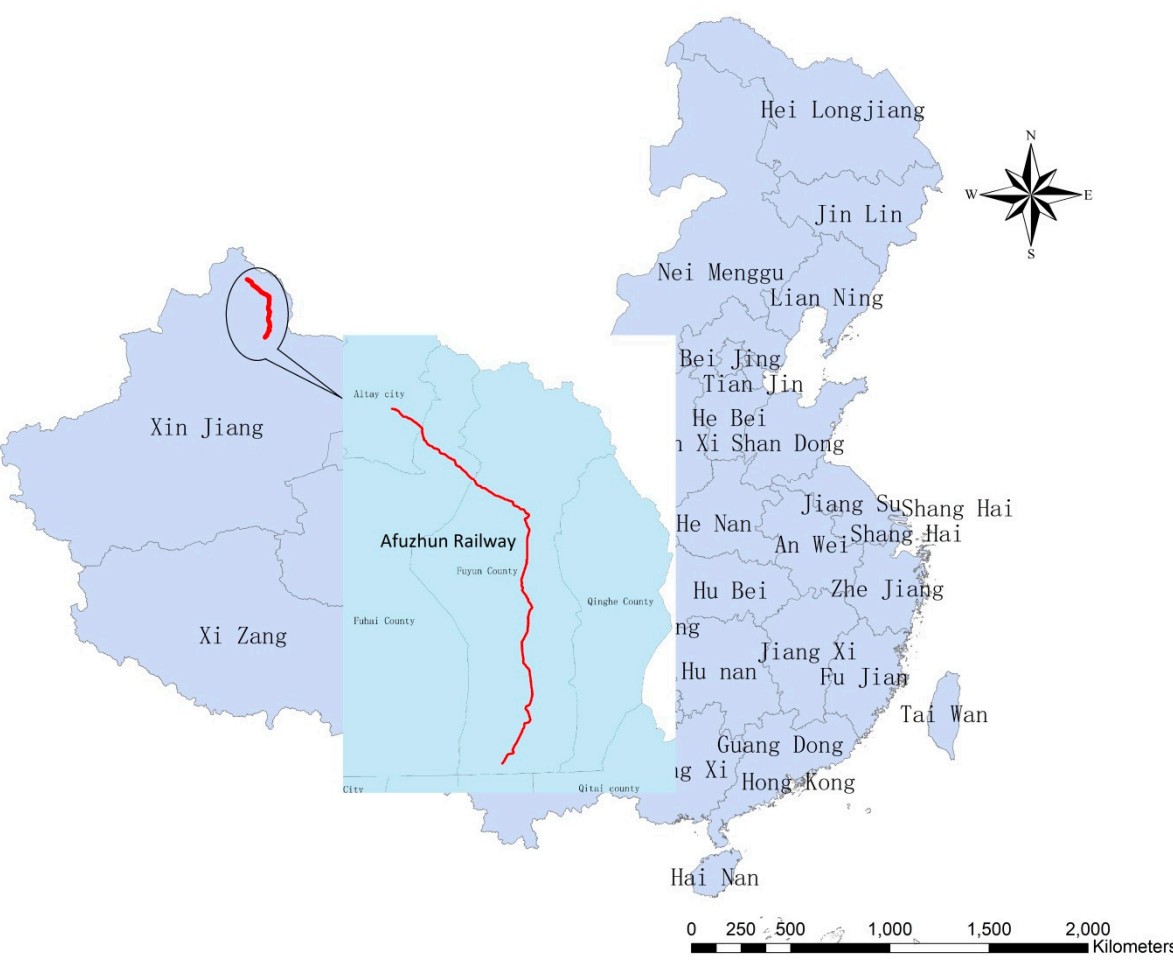

**Figure 3.** Route map.

*3.2. Data Source*

The susceptibility zone based on GIS requires a large amount of raster data, including terrain data, wind field data, and snow depth data along the line. Therefore, the research data in this study is divided into three categories, and the raster resolution is 10 m × 10 m:

1.    Terrain data

Topographic data include elevation data, topographic relief, roughness, and slope aspect. Based on the 30M resolution digital elevation data of the "Geospatial Data Cloud", this part of the data can be obtained using GIS-related analysis functions.

2.    Wind field data

The wind speed and dominant wind direction determine the direction and trend of snow flow movement. The AFZ Railway is located in a mountainous area with complex terrain, which significantly changes the wind speed and wind direction. The research requirements cannot be met only by relying on historical meteorological data of nearby weather stations. Therefore, for two consecutive years, 20 meteorological monitoring stations (Figure 4), set up along the entire railway line, monitored the wind speed and direction in the project area.

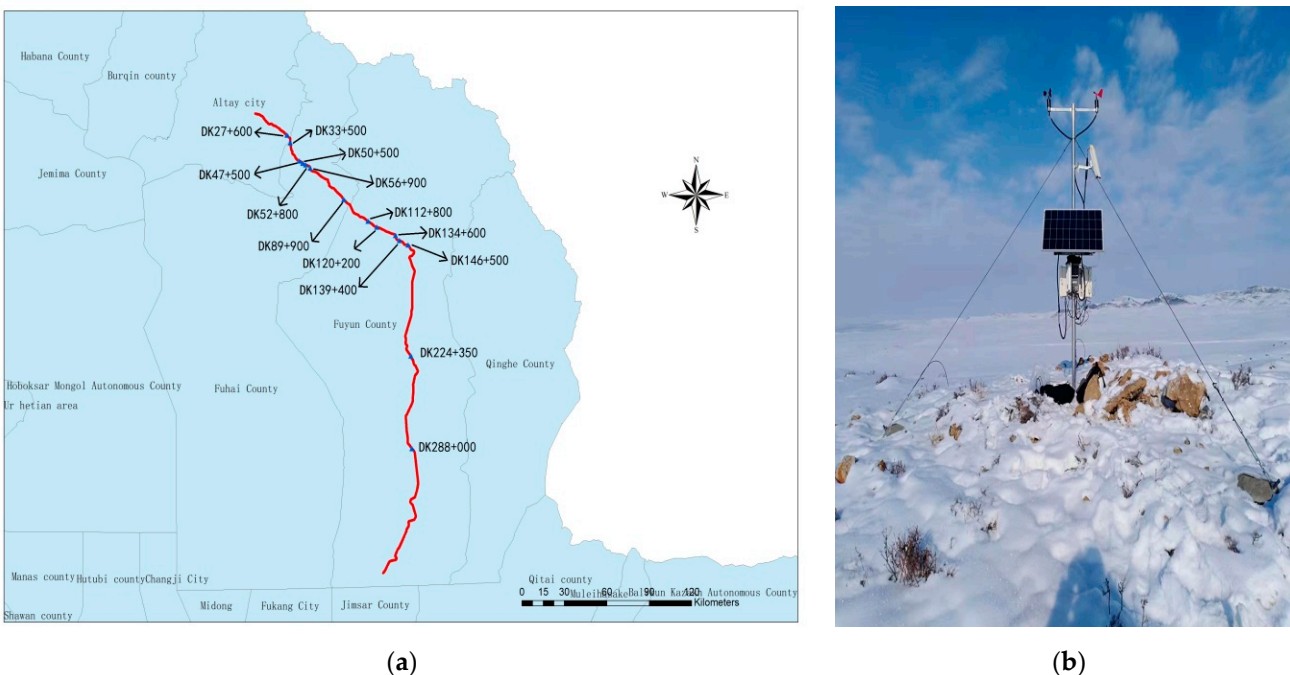

(**a**)　　　　　　　　　　　　　　　　　(**b**)

**Figure 4.** Monitoring points and weather stations. (**a**) Layout of monitoring points, (**b**) Weather stations.

3.　On-site snow depth data

The thickness of the drifting snow along the railway is an important standard to measure its degree of severity and evaluate the susceptibility of drifting snow. Therefore, our research group organised personnel to conduct field measurements of the thickness of the drifting snow along the railway (Figure 5) and established the corresponding data set.

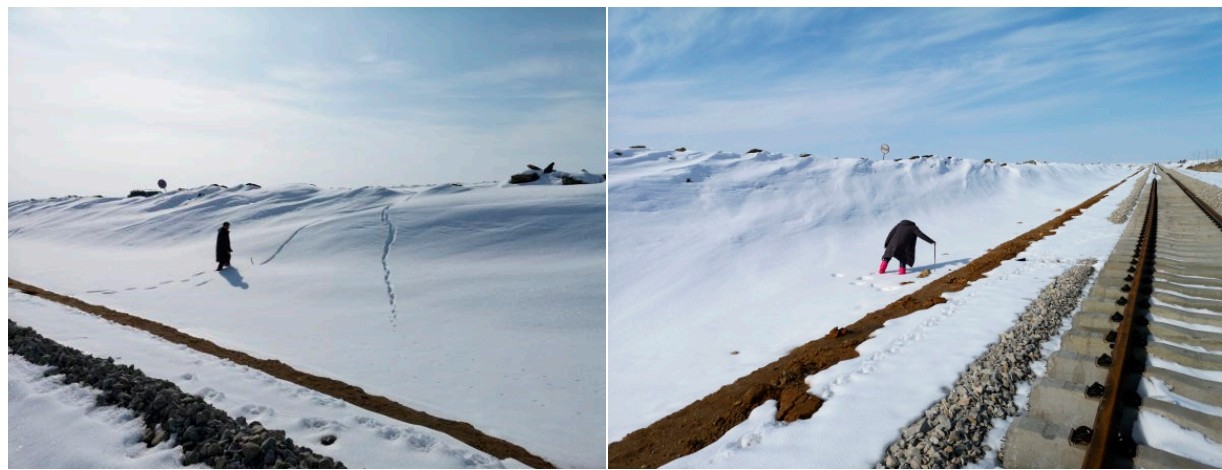

**Figure 5.** Monitoring points and weather stations.

## 4. Establishment of Evaluation Index System and Model Construction

### 4.1. Evaluation Index System

The occurrence of drifting snow disasters is the result of natural snowfall, wind field, and terrain factors. Based on previous research results and field data from regional environmental conditions, that is, regional snowfield and wind field conditions, this study selected elevation, relief amplitude, surface roughness, aspect, snowfall, frequency of heavy snowfall, average wind speed, maximum wind speed, the angle between the wind and the line directions, and snow wind speed frequency. Therefore, a total of ten influencing factors were used to build the evaluation index system (Figure 6).

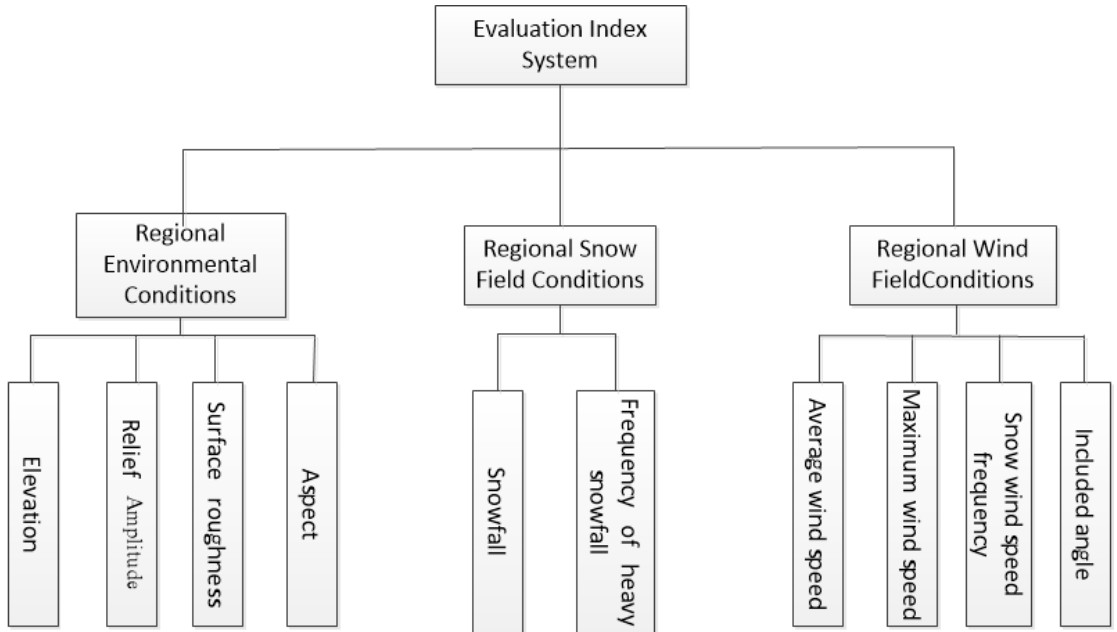

**Figure 6.** Evaluation index System.

*4.2. Index Factor Classification*

Index factor data includes both continuous and discrete types of data. For the continuous data, the physical meaning needs to be discretised. For discrete data, each level has a clear physical meaning. Finally, the drifting snow area ratio, graded area ratio, and WOE are used to comprehensively evaluate the impact of the secondary status of each index factor on the drifting snow [50]. The drifting snow area ratio is equal to the area of the drifting snow in the secondary state of the indicator factor/the total area of the drifting snow in the whole region; the graded area ratio is the area of each secondary state of the index factor compared to the total area of the index factor. Their relative sizes represent the importance of each second-level state classification of this indicator factor to the drifting snow susceptibility [51]. If the area ratio of the drifting snow is greater than that of the graded area ratio, it indicates that drifting snow can easily occur in the graded state; otherwise, the opposite is true.

4.2.1. Regional Environmental Conditions

1. Elevation

The influence of altitude on drifting snow is reflected in two aspects. Firstly, altitude can affect the wind speed and the size of the flow section, that is, the flow section decreases with the increasing altitude, and the wind speed increases in high altitude areas; Secondly, the wind speed can affect the temperature; moreover, the size of snow particles has a close relationship with the temperature. The temperature can affect the melting speed of snow particles. The deeper the melting degree of snow particles, the larger is the particle size and the faster the corresponding starting wind speed. The elevation of this study area is between 597 m and 1219 m, the occurrence of drifting is concentrated between 597 m and 1063 m, and there is no drifting snow over 1063 m. According to the actual distribution, the elevation is divided into five secondary states: 597–764 m, 764–865 m, 865–955 m, 955–1063 m, and 1063–1219 m. The area ratio of drifting snow, the area ratio of classification, and the WOE in each secondary state were calculated (Figure 7).

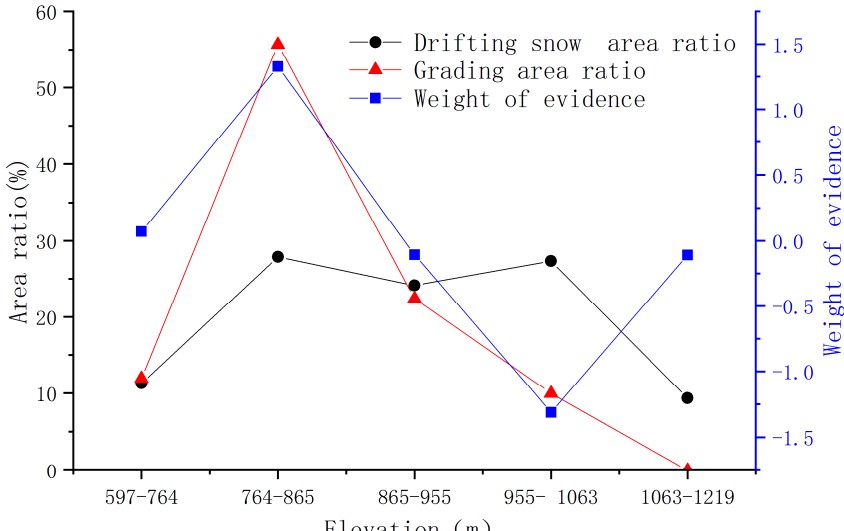

**Figure 7.** Elevation statistical analysis results.

2.  Relief amplitude

Relief amplitude refers to the difference in elevation between the highest point and the lowest point in a certain range, indicating the steepness of the terrain in a certain area. The size of the relief amplitude can also reflect the degree of irregularity of the ground in a certain area. When the relief amplitude is small, the near-ground wind movement is affected by friction resistance, and the wind speed changes slightly. The ground snow particles are affected by a horizontal force, but the vertical force is not noticeable, and the snow particles have no upward movement trend. Moreover, when the relief amplitude is large, air turbulence becomes larger and a vortex is formed. At this time, the vertical force on snow particles on the ground increases. When the vertical force is greater than the gravitation force on snow particles, supplemented by transverse wind, snow particles fly from the ground and form drifting snow. Therefore, in this study, the relief amplitude is divided into five secondary states: 0–17, 17–25, 25–37, 37–59, 59–137. The area ratio of drifting snow, the area ratio of classification, and the WOE in each secondary state were counted (Figure 8).

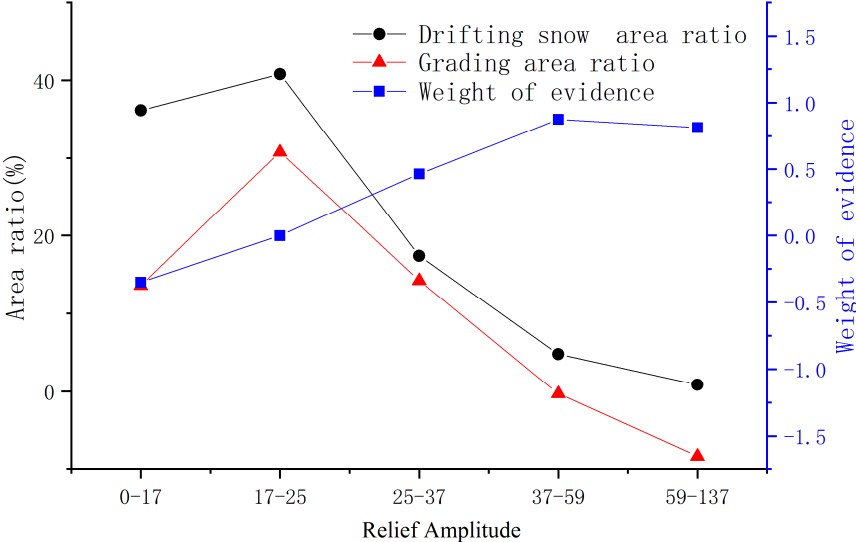

**Figure 8.** Elevation statistical analysis results.

3.    Surface roughness

Surface roughness is the ratio of the actual ground surface to the projected area within a certain range, reflecting the roughness of the ground surface. The formula is as follows:

$$SR = (AC.AB)/(AC.AC) = 1/\cos a \tag{3}$$

In the formula, *a* is the slope of the secondary grid unit; then, the area of the *AB* surface is the surface area of the secondary grid, and the area of the *AC* surface is the projected area of the grid, where cos *a* = *AC/AB*, with the result being the secant of the slope.

Relief amplitude affects fluid turbulence over relatively large distances, whereas surface roughness affects turbulence only over small distances. However, both these factors can increase the vertical force on snow particles; moreover, it is easy to move under the action of transverse force and finally form drifting snow. Therefore, in this study, the surface roughness is divided into four secondary states: 1–1.009, 1.009–1.034, 1.034–1.12, and 1.12–1.668. The area ratio of drifting snow, the area ratio of classification, and the WOE in each secondary state were calculated (Figure 9).

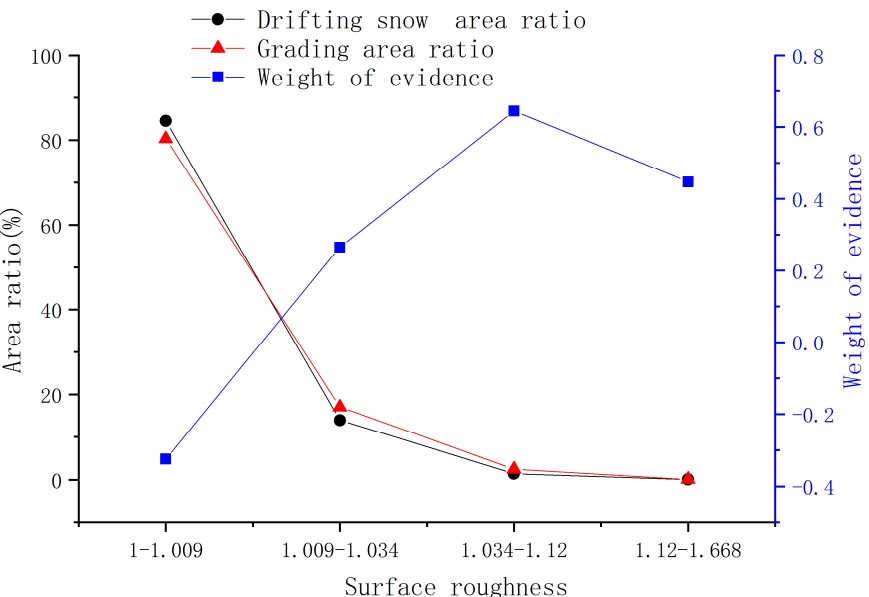

**Figure 9.** Surface roughness statistical analysis results.

4.    Aspect

The difference in the aspect of the slope leads to the different times of the sun. A long sunshine time results in a sunny slope; short sunshine time leads to a shaded slope. The southern slope of the northern hemisphere has a longer sunshine time than the north slope. The impact of slope direction is reflected in two aspects. Firstly, different slope directions receive different light hours, resulting in different regional temperatures, which affect the melting speed of snow particles [52]. Secondly, there are differences in vegetation in different slope directions. Plants on sunny slopes grow more abundantly, and their aboveground crown and root systems are more developed, meaning they also have a better blocking effect on drifting snow. Hence, in this study, the aspect is divided into six secondary states: 0–60°, 60–120°, 120–180°, 180–240°, and 240–360°. The area ratio of drifting snow, the area ratio of classification, and the WOE in each secondary state were calculated (Figure 10).

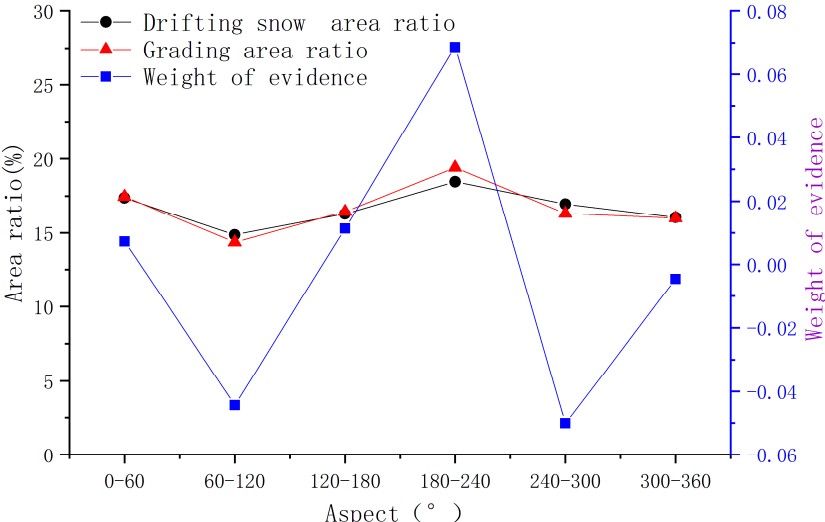

**Figure 10.** Aspect statistical analysis results.

### 4.2.2. Regional Snow Field Conditions

1. Snowfall

The annual snowfall is an important factor in drifting snow, and the snow source in a certain area is termed sufficient based on the amount of snowfall. According to the snowfall data of the "Geospatial Data Cloud" platform, the snowfall in the study area is divided into four secondary states: 40–900 mm, 900–1350 mm, 1350–1700 mm, and 1700–1928 mm. The area ratio of classification and the WOE in each secondary state were calculated (Figure 11).

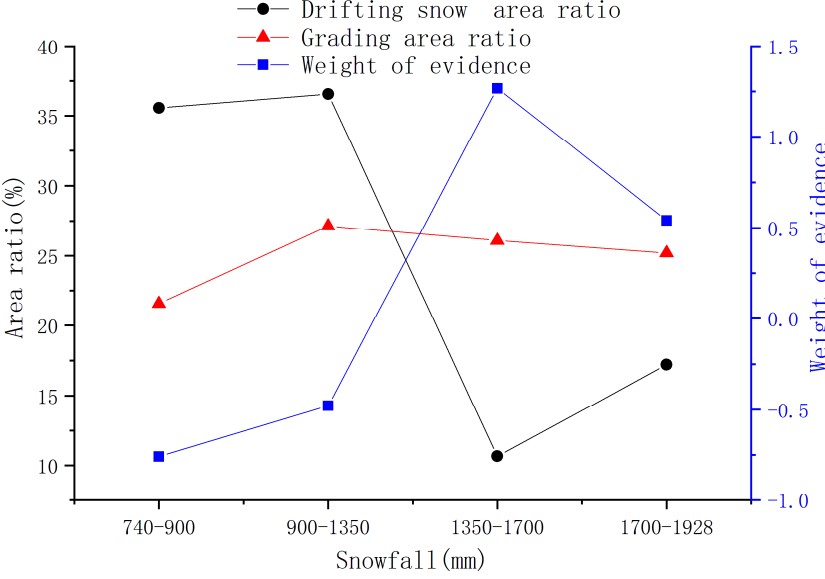

**Figure 11.** Snowfall statistical analysis results.

2. Frequency of heavy snowfall

According to different times, the snow particles on the ground can be divided into new and old snow. The particle size of new snow is relatively smaller than that of old snow. There is a high incidence of drifting snow, usually within a few days after snowfall. The longer the time since the snowfall, the larger is the size of the snow particles; therefore, the occurrence of drifting snow is related to the frequency of snowfall. The more frequent the snowfall, the greater is the total amount of fresh snow and the greater the possibility of drifting snow. According to previous research results and the statistical data of the National

Meteorological Station on Altay region (1961–2013) [53], the frequency of heavy snowfall in this study area can be divided into four secondary states: 1.5–2.0, 2.05–2.75, 2.75–3.1, and 10–3.50. The area ratio of classification and the WOE in each secondary state were calculated (Figure 12).

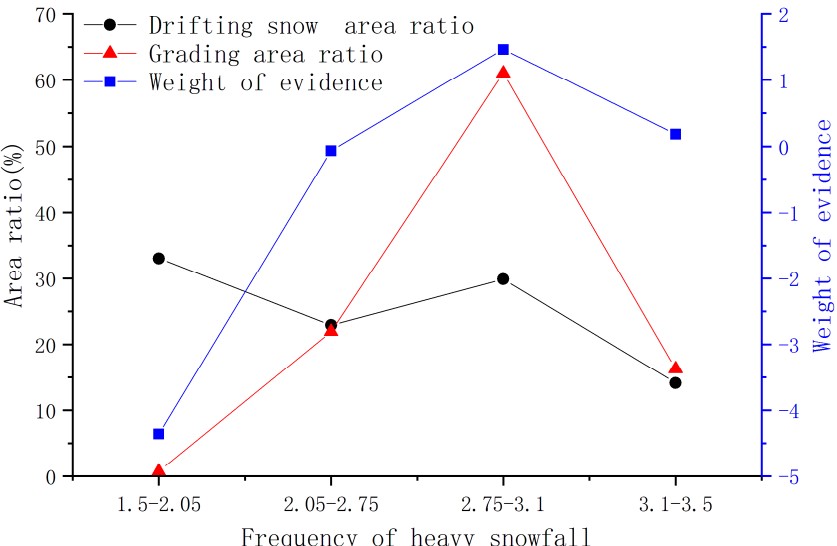

**Figure 12.** Frequency of heavy snowfall statistical analysis results.

4.2.3. Regional Wind Field Conditions

1. Wind speed

The wind is a necessary condition for the formation of drifting snow. According to the mechanical formula, the movement of snow particles is a function of wind speed. Therefore, wind speed is an important factor in the evaluation of drifting snow susceptibility.

There are many statistical analysis methods for wind speed. For example, the average and maximum wind speed are typical statistical methods. To further analyse the wind speed data collected on-site, the wind speed statistics data are divided into two types: the average and maximum wind speeds; both of them are used as drifting snow evaluation indicators. According to the wind speed data collected on-site in the winter of 2018 and 2019 (Table 1), the average wind speed and maximum wind speed are divided into four secondary states. The average wind speed is divided into 1.29–1.34 m/s, 1.34–1.67 m/s, 1.67–1.8 m/s, and 1.8–2.73 m/s states, and the maximum wind speed is divided into 7.7–11.7 m/s, 11.7–13.05 m/s, 13.05–13.7 m/s, and 13.7–15.25 m/s states. The area ratio of classification and the WOE in each secondary state were calculated (Figure 13).

**Table 1.** Statistics of average wind speed and maximum wind speed along the line.

| Monitoring Mileage | Average Wind Speed (m/s) | Maximum Wind Speed (m/s) | Monitoring Mileage | Average Wind Speed (m/s) | Maximum Wind Speed (m/s) |
|---|---|---|---|---|---|
| DK89+900 | 1.5 | 14.1 | DK89+900 | 1.76 | 12.5 |
| DK112+800 | 1.45 | 7.8 | DK112+800 | 1.23 | 15.6 |
| DK120+200 | 1.68 | 12.8 | DK120+200 | 1.65 | 13.7 |
| DK129+300 | 1.79 | 11.8 | DK129+300 | 1.32 | 13 |
| DK134+600 | 2.89 | 13.4 | DK134+600 | 1.45 | 14 |
| DK139+400 | 1.32 | 10.6 | DK139+400 | 1.26 | 19.9 |
| DK146+500 | 2.62 | 17.2 | DK146+500 | 0.98 | 7.7 |
| DK224+350 | 2.74 | 11.7 | DK224+350 | 1.64 | 13.5 |
| DK288+000 | 2.63 | 12.8 | DK288+000 | 2.82 | 13.3 |
| DK322+150 | 1.64 | 12.1 | DK322+150 | 2.8 | 13.7 |

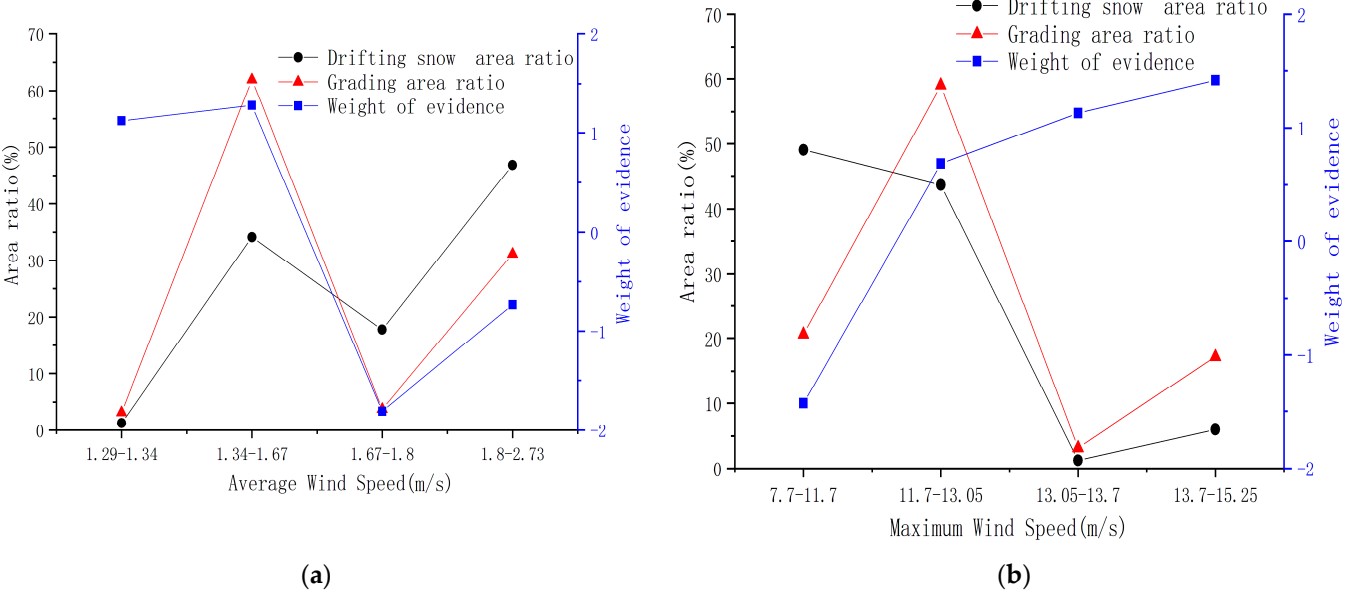

(**a**)                                               (**b**)

**Figure 13.** Wind speed statistical analysis results. (**a**) Average wind speed, (**b**) Maximum wind speed.

2.      Angle between wind and line direction

The angle between the wind and line directions can greatly affect the snow resistance of regional road sections caused by drifting snow. The line and wind directions change between 0° and 90°. When the angle between the line and the wind direction is 90°, the line has the greatest blocking effect on wind and snow flow, and snow particles form a great accumulation on the line's subgrade. When the angle between the line and the wind direction is 0°, the snow particles in the drifting snow flow are consistent with the line direction and are not blocked by the line but accumulate on the line roadbed. According to the monitoring data obtained in the field and combined with the actual situation in the research area, the angle between the main wind and line directions is divided into five secondary states: 0–17.5°, 17.5–6.5, 36.5–55.5°, 55.5–74.5°, and 74.5–89°. The area ratio of classification and the WOE in each secondary state were calculated (Figure 14).

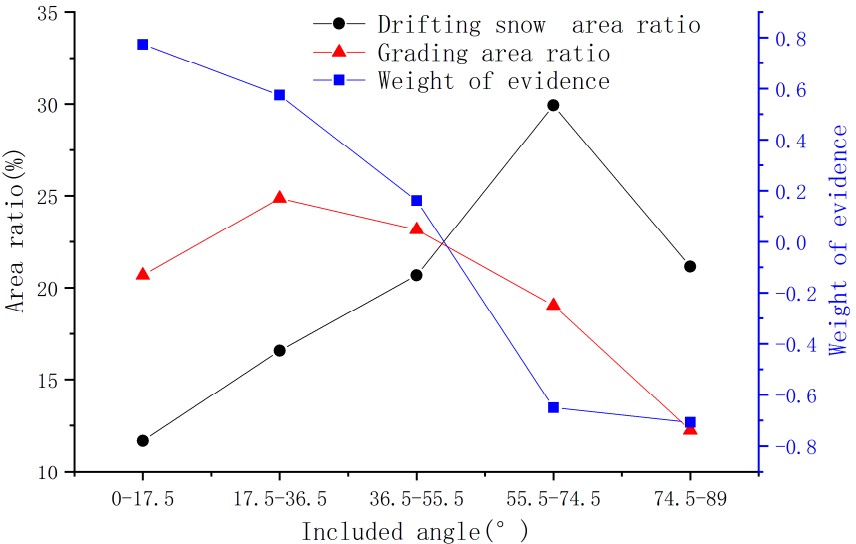

**Figure 14.** Included angle statistical analysis results.

3. Snow wind speed frequency

The starting wind speed of snow particles is greatly related to the size of the snow particles. Generally, the starting wind speed of snow particles with small particle sizes is lower. According to research findings, when the temperature is below −6 °C, the snowfall starting wind speed is approximately 3.4 m/s [54]. The winter temperature in the study area is relatively low. In most cases, the temperature is lower than −6 °C. The starting snow wind speed measured by the on-site wind tunnel test is approximately 4 m/s (3 m high). Therefore, the speed of 4 m/s is defined as the starting snow wind speed, and the statistical results are shown in Table 2. The frequency of snow wind speed in the study area is divided into six secondary states: 0.55–1.59%, 1.59–4.58%, 4.58–6.06%, 6.06–12.2%, 12.2–17.7%, 17.7–23.18%. The area ratio of classification and the WOE in each secondary state were calculated (Figure 15).

**Table 2.** Statistic table of snow wind speed frequency.

| Monitoring Mileage | Wind Speed Frequency above 4 m/s | Monitoring Mileage | Wind Speed Frequency above 4 m/s |
|---|---|---|---|
| DK27+600 | 4.42% | DK89+900 | 4.58% |
| DK33+500 | 6.06% | DK112+800 | 4.45% |
| DK42+800 | 12.20% | DK120+200 | 3.01% |
| DK44+700 | 21.67% | DK129+300 | 1.59% |
| DK47+500 | 22.48% | DK134+600 | 4.52% |
| DK50+500 | 9.17% | DK139+400 | 5.23% |
| DK52+800 | 21.06% | DK146+500 | 0.55% |
| DK56+900 | 22.81% | DK224+350 | 10.64% |
| DK62+600 | 17.71% | DK288+000 | 23.18% |

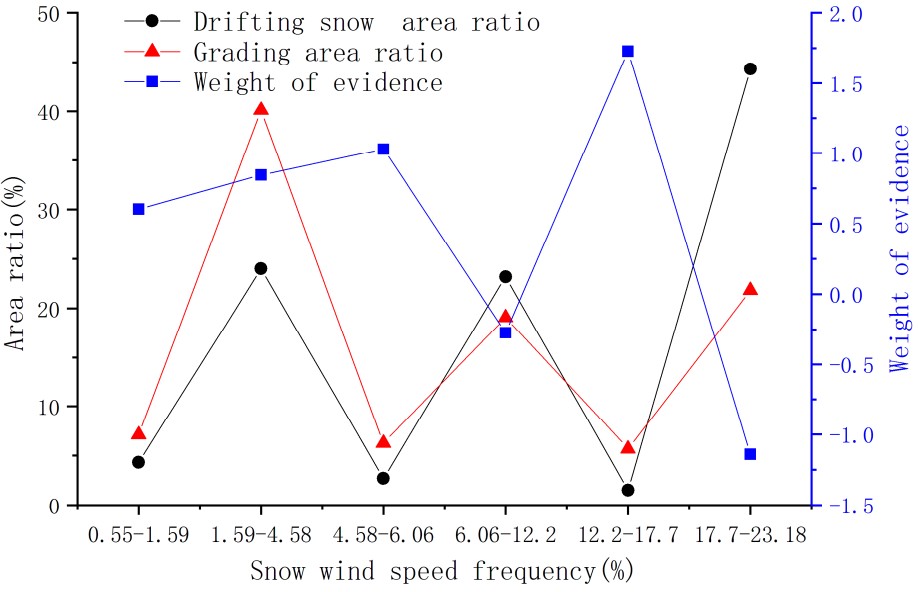

**Figure 15.** Snow wind speed frequency statistical analysis results.

*4.3. Analysis of the Construction of WOE Model Improved by GA-BP Algorithm*

4.3.1. Weight of Evidence (WOE) Model

We used the formulae in Section 2.1 (Equations (1) and (2)) to calculate the evidence weight of each indicator's secondary status factor (Table 3). On this basis, ArcGIS was used to superimpose each impact factor layer to calculate the drifting snow susceptibility index (DSSI) of the entire study area. DSSI is the algebraic sum of $W_{fi}$ calculated using the grid

calculator overlay, and its numerical value can represent the degree of susceptibility to drifting snow. The DSSI in the study area is in the range −8.27–9.89.

$$\text{DSSI} = \sum_{i=1}^{n} W_{fi} \qquad (4)$$

**Table 3.** Index factors and $W_{fi}$.

| Index Factors | Index Classification | $W_{fi}$ | Index Factors | Index Classification | $W_{fi}$ |
|---|---|---|---|---|---|
| Elevation (m) | 597–764 | 0.071 | Average wind speed (m/s) | 1.29–1.34 | 1.121 |
|  | 764–865 | 1.3275 |  | 1.34–1.67 | 1.277 |
|  | 865–955 | −0.108 |  | 1.67–1.8 | −1.808 |
|  | 955–1063 | −1.312 |  | 1.8–2.73 | −0.732 |
|  | 1063–1219 | −0.109 | Maximum wind speed (m/s) | 7.7–11.7 | −1.4303 |
| Relief Amplitude | 0–17 | −0.350 |  | 11.7–13.05 | 0.682 |
|  | 17–25 | −0.002 |  | 13.05–13.7 | 1.132 |
|  | 25–37 | 0.465 |  | 13.7–15.25 | 1.417 |
|  | 37–59 | 0.876 | Included angle (°) | 0–17.5 | 0.774 |
|  | 59–137 | 0.815 |  | 17.5–36.5 | 0.575 |
| Aspect (°) | 0–60 | 0.007 |  | 36.5–55.5 | 0.161 |
|  | 60–120 | −0.044 |  | 55.5–74.5 | −0.650 |
|  | 120–180 | 0.011 | Snow wind speed frequency (%) | 0.55–1.59 | 0.603 |
|  | 180–240 | 0.069 |  | 1.59–4.58 | 0.8443 |
|  | 240–300 | −0.050 |  | 4.58–6.06 | 1.029 |
|  | 300–360 | −0.005 |  | 6.06–12.2 | −0.273 |
| Surface roughness | 1–1.009 | −0.325 |  | 12.2–17.7 | 1.730 |
|  | 1.009–1.034 | 0.265 |  | 17.7–23.18 | −1.142 |
|  | 1.034–1.12 | 0.645 | Frequency of heavy snowfall (%) | 1.5–2.05 | −4.3617 |
|  | 1.12–1.668 | 0.447 |  | 2.05–2.75 | −0.062 |
| Snowfall(mm) | 740–900 | −0.759 |  | 2.75–3.1 | 1.463 |
|  | 900–1350 | −0.479 |  | 3.1–3.5 | 0.185 |
|  | 1350–1700 | 1.266 |  |  |  |
|  | 1700–1928 | 0.541 |  |  |  |

$W_{fi}$—Weight of evidence for each secondary state factor.

There are 22,912,695 grids in the area along the railway, including 1,405,522 grids for low-prone areas, 4869708 grids for medium-prone areas, and 3988465 grids for high-prone areas, accounting for 61.34%, 21.25%, and 17.41%. High-susceptibility areas are concentrated in DK57–DK116 Km, moderately susceptible areas are concentrated in DK0–DK57 Km, and DK116–DK139 Km, and the rest are low-prone areas (Figure 16).

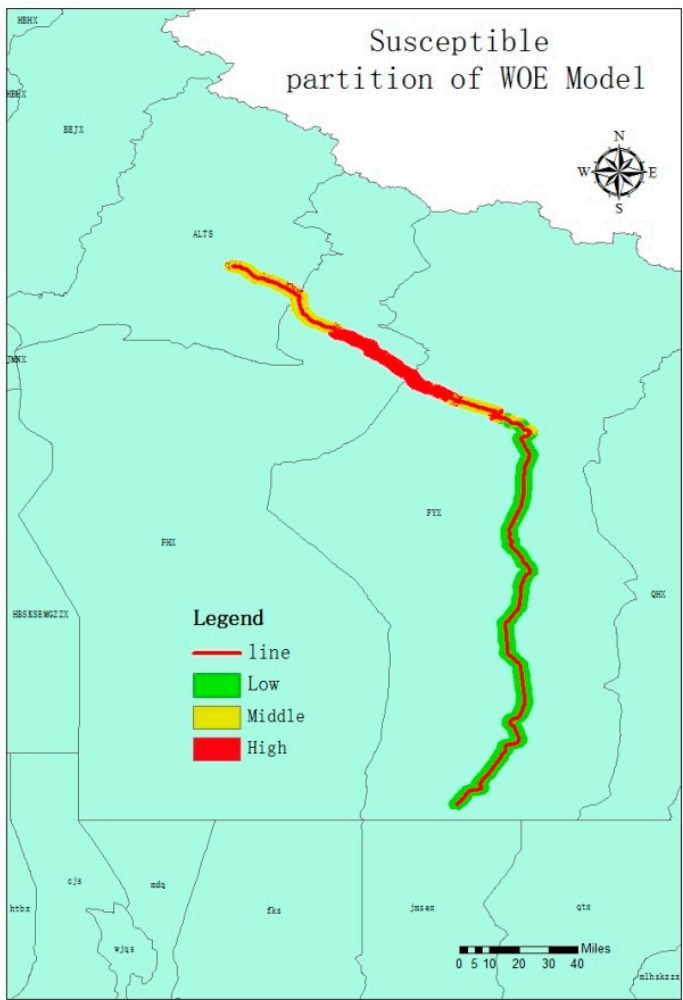

**Figure 16.** Susceptible partition of WOE Model.

### 4.3.2. GA-BP Algorithm Optimisation

Although the WOE model can characterise the impact of each index factor on the risk, it does not consider the weight ratio of each factor in the stacking calculation, which ignores the relative importance of each index factor to the overall contribution. However, the neural network model has noticeable black-box characteristics and can mine nonlinear behaviours from training data, which has notable advantages in the weight determination of evaluation factors [55–57]. The genetic algorithm can optimise the weights of the neural network and improve the evaluation accuracy of the model. The construction of the neural network model has been studied in detail in numerous research [58–60] and is, therefore, not mentioned here. This study constructs a neural network model of 10–12–1. The model contains ten input layer neurons corresponding to the evaluation indicators one by one, one output layer neuron corresponding to the DSSI, and 12 hidden layer neurons.

There are 22,912,695 grid units in this research area and 2,171,375 grid units in the drifting snow area. We randomly selected 2000 grids from the drifting snow grid and the non-drifting snow grid, a total of 4000 grid cells. The evaluation index attributing data corresponding to each grid is extracted as the input layer, and the DSSI is extracted as the output layer to construct neural network training samples. First, we used the BP neural network for training, calculated the weight value, and then used the GA algorithm to improve the neural network (Figure 17), which was used for retraining and to calculate the weight value (Table 4).

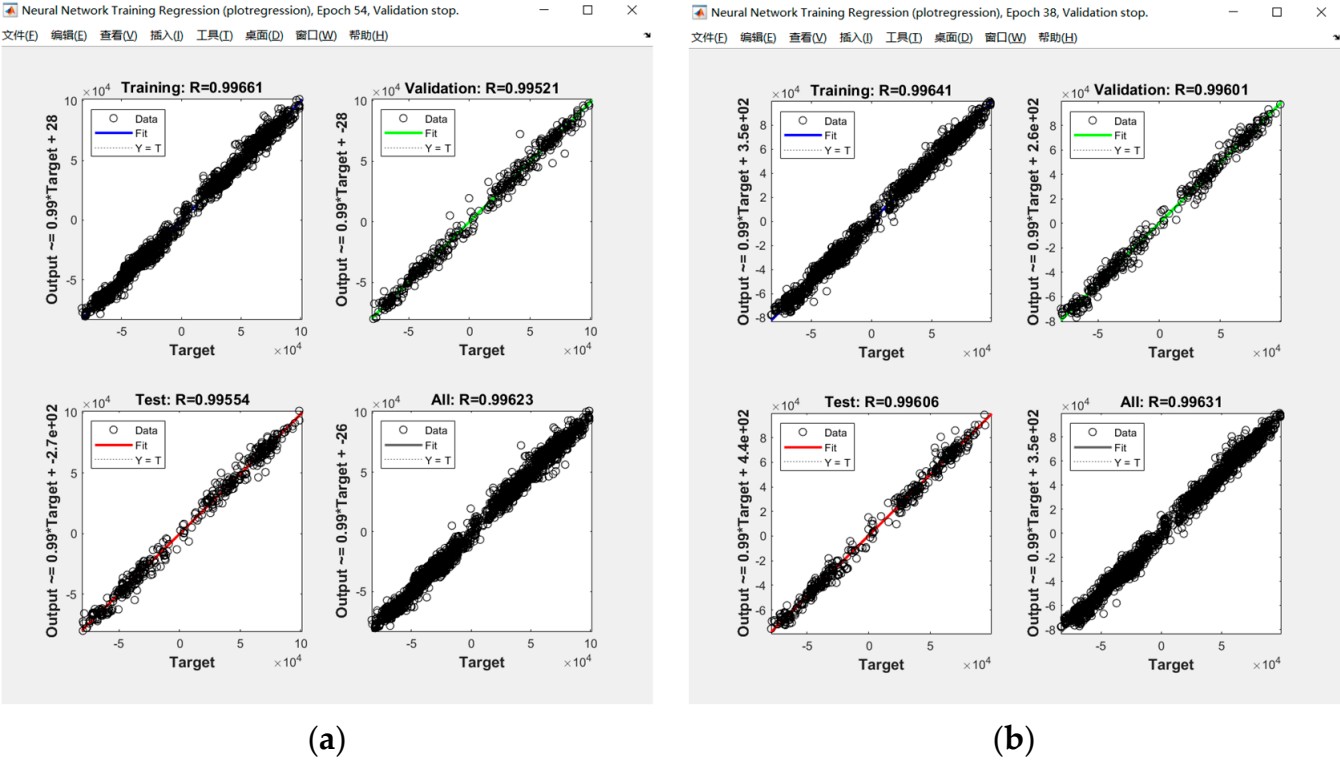

**Figure 17.** Correlation coefficients of neural network training. (**a**) BP, (**b**) GA-BP.

**Table 4.** Calculated index factor weights.

| Serial Number | Index Factor | BP Weight | GA-BP Weight |
|:---:|:---:|:---:|:---:|
| 1 | Elevation | 0.880037 | 0.870972 |
| 2 | Relief Amplitude | 1.129359 | 0.799087 |
| 3 | Aspect | 0.815339 | 0.663054 |
| 4 | Surface roughness | 0.744768 | 0.275845 |
| 5 | Snowfall | 0.763512 | 0.651298 |
| 6 | Frequency of heavy snowfall | 1.134141 | 1.399246 |
| 7 | Average wind speed | 0.993594 | 1.684908 |
| 8 | Maximum wind speed | 1.116979 | 1.915338 |
| 9 | Included angle | 1.473144 | 2.706506 |
| 10 | Snow wind speed frequency | 0.949127 | 0.933745 |

Figure 17 shows that using a BP neural network or a GA-BP neural network, the correlation coefficient R, after training, is above 0.99, which is consistent with the data. Through weight calculation, the factors with a relatively large contribution to drifting snow calculated using both training models are the same. The highest weight is the angle between the wind and the line directions, followed by the maximum wind speed; the weight of snowfall frequency is also large, which also confirms the importance of dynamic factors and the provenance of drifting snow.

**5. Drifting Snow Evaluation and Accuracy Analysis Susceptibility Evaluation**

*5.1. Susceptibility Evaluation*

Based on the WOE model calculation and ArcGIS grid calculator, the weight results in Table 4 were substituted into the optimisation calculation, and an optimised drifting snow susceptibility zoning diagram (Figure 18) could be reconstructed. The results show that

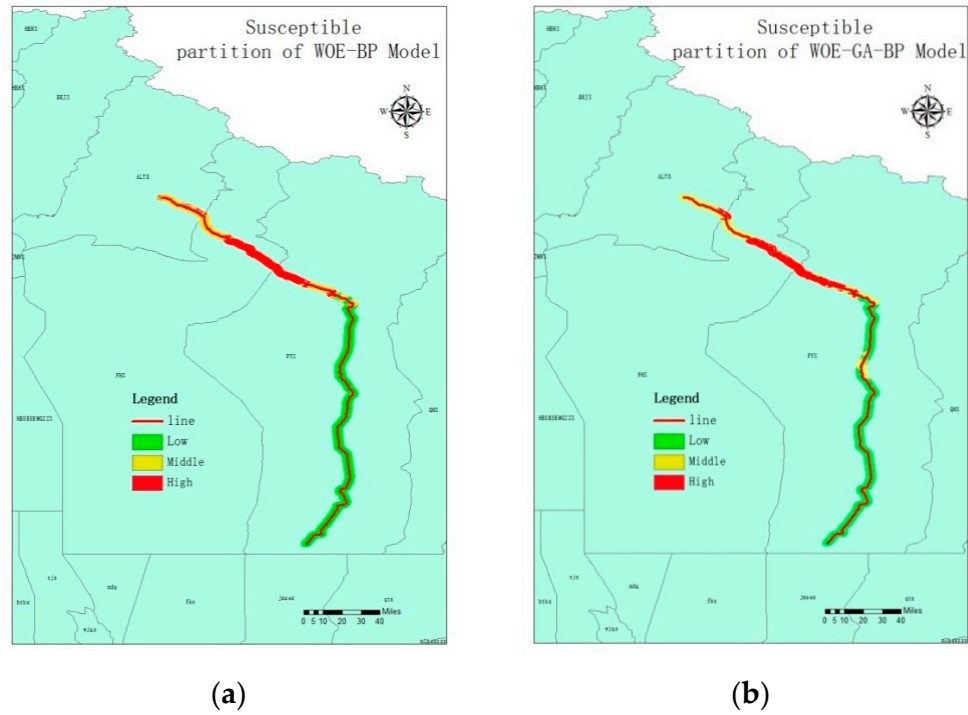

(**a**)                                (**b**)

**Figure 18.** Hazard partition. (**a**) WOE-BP Model, (**b**) WOE-GA-BP Model.

1.  The location of the high-susceptibility concentration area calculated by the three models is unchanged, and all of them are between DK57 km and DK116 km. After the BP neural network weight optimisation, the DK 230–DK 231+300 km susceptibility grade is improved, from low susceptibility area to medium susceptibility area. After the GA-BP model weight optimisation, the susceptibility level of DK23–DK33+400 km is improved, especially the DK32+400 km–DK33+400 km section is changed from a medium- to a high-prone area.
2.  The weight analysis results are consistent with the theoretical analysis. The factors that have a greater contribution to the drifting snow susceptibility are the angle between the wind and the line directions, the maximum wind speed, the amount of snowfall, and the frequency of snowfall.
3.  Compared with the WOE and WOE-BP models, the WOE-GA-BP model obtained a higher proportion of high-susceptibility areas in the susceptibility zoning map and included more drifting snow occurrence areas, which has more practical significance for guiding drifting snow protection areas.

*5.2. Accuracy Evaluation*

In this study, the receiver operating characteristic (ROC) curve and area under the curve (AUC) were taken as the measurement standards. Fawcett conducted a detailed study on the basic theory and calculation method of the ROC curve and AUC [61].

According to the research results, the corresponding results for the AUC value and evaluation accuracy are as follows: AUC < 0.7, poor evaluation accuracy; 0.7 < AUC < 0.8, medium evaluation accuracy; AUC > 0.8, good evaluation accuracy. This study divides the DSSI into 100 intervals from large to small and gradually counts the cumulative occurrence frequency of drifting snow in each interval. The total grid frequency of the study area

is taken as the abscissa, and the cumulative occurrence frequency of drifting snow is the vertical coordinate when drawing the ROC curve (Figure 19).

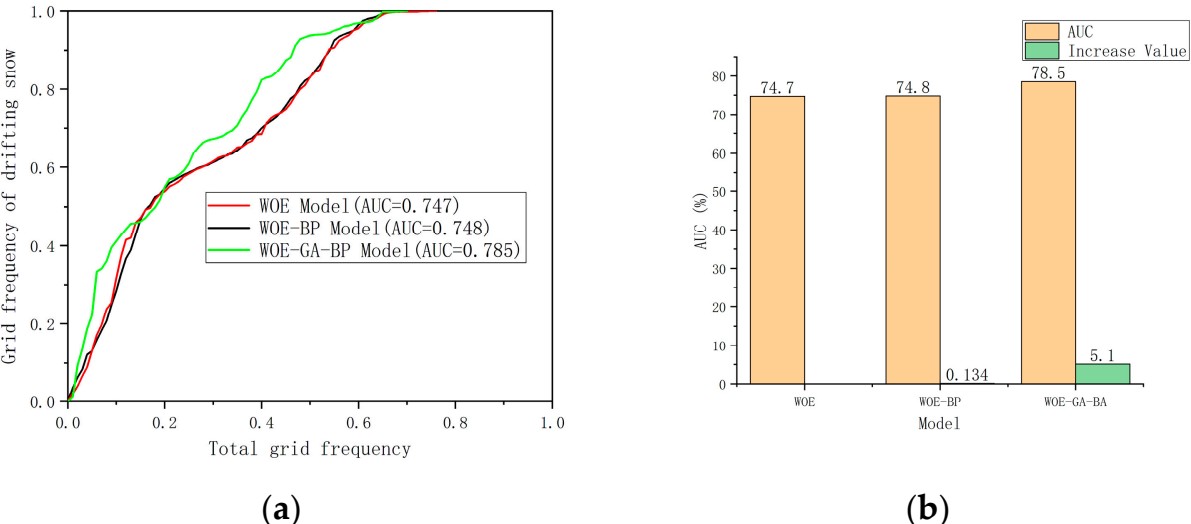

**Figure 19.** ROC curve and accuracy comparison chart. (**a**) ROC curve, (**b**) Comparison chart.

According to the ROC curve, the AUC areas corresponding to the three evaluation models are 0.747 (WOE), 0.748 (WOE-BP), and 0.785 (WOE-GA-BP), all of which have achieved good evaluation accuracy. After the BP algorithm optimises the weights, the model evaluation accuracy is improved by only 0.134%. After the GA-BP algorithm optimises the weights, the model evaluation accuracy is considerably improved, reaching 5.1%, which is close to the optimal standard of 0.8.

## 6. Conclusions

In this paper, the WOE model, the WOE-BP model, and the WOE-GA-BP model were used to study the evaluation index system of drifting snow susceptibility along railways. The influence of the weight of each index and the evaluation effect of each model were combined with the wind field data obtained from field monitoring. The results indicated the following:

1.  Taking Afuzhun Railway in Xinjiang as the specific research object, the evaluation index system for drifting snow susceptibility on the railway can be established by selecting ten influencing factors, such as elevation, snowfall, the angle between the wind and the line directions. Using the DSSI proposed in this paper, the WOE model can be used to obtain the preliminary zoning of drifting snow susceptibility along the railway.

2.  Based on the initial classification of drifting snow susceptibility, 4000 grid cells were randomly selected as training samples, and the influence of each index on the weight of the evaluation results was calculated using the BP and GA-BP algorithms. The results showed that the weight of the angle between the wind and the line directions was the largest, followed by the largest wind speed and the frequency of heavy snowfall.

3.  The calculated weights were used to optimise the WOE model. The results showed that the evaluation accuracy of all models was improved. The GA-BP algorithm improved the evaluation accuracy by 5.1% to 0.785, achieving a high evaluation accuracy.

4.  The GA-BP algorithm can effectively study the complex nonlinear relationship between various indicators and obtain evaluation results that are highly consistent with the actual situation. This method can effectively find the high incidence area of drifting snow in linear railways and provide a theoretical basis for the effective prevention and control of drifting snow.

5.  In the relatively mature application fields for WOE, such as landslide susceptibility evaluation, the evaluation accuracy of this evaluation model can reach 0.8 or even higher. In comparison, the evaluation accuracy of this model for linear engineering (such as railway) is low. In the future application of this research, we will focus on optimising the evaluation model to improve the evaluation accuracy of the model.

6.  The evaluation accuracy of this method partly depends on the accuracy of data, especially the wind field data. This study is based on on-site monitoring and takes considerable time to collect data. Hence, further research is required to develop a more efficient and simple method to obtain field data.

**Author Contributions:** Conceptualization, Bohu He and Mingzhou Bai; methodology, Bohu He; software, Bohu He; validation, Binglong Liu, Bohu He and Pengxiang Li; formal analysis, Lusheng Ding; investigation, Binglong Liu, Pengxiang Li, Shumao Qiu and Xin Li; resources, Lusheng Ding; data curation, Binglong Liu; writing—original draft preparation, Bohu He; writing—review and editing, Mingzhou Bai; visualization, Bohu He; supervision, Mingzhou Bai and Lusheng Ding; project administration, Mingzhou Bai and Lusheng Ding; funding acquisition, Mingzhou Bai. All authors have read and agreed to the published version of the manuscript.

**Funding:** This research was funded by the National Key Research and Development Project of China, the Ministry of Science and Technology of China (grant 2018YFC1505504), and Fundamental Research Funds for the Central Universities (2020YJS119).

**Data Availability Statement:** Data are available upon reasonable request to the corresponding authors.

**Conflicts of Interest:** The authors declare no conflict of interest.

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
