# Peer review of "Evaluation of Drifting Snow Susceptibility Based on GIS and GA-BP Algorithms"

_ijgi, doi:10.3390/ijgi11020142_

Round 1
Reviewer 1 Report
The manuscript “Evaluation of Drifting Snow Susceptibility based on GIS and GA-BP algorithms)” by He et al. combines meteorological observations and GIS modeling to compute a drifting snow susceptibility index to evaluate the susceptibility of the Xinjian Afukuzhun Railroad to drifting snow. The GIS model employs both a weight-of-evidence (WOE) and genetic algorithm backpropagation to develop the model.
Overall, the research appears sound and the results are reasonable as is the interpretation. However, in its current form the manuscript suffers from some issues with clarity. It is difficult to understand some of the steps in the process the authors use to arrive at their conclusions. With these minor-to-moderate changes the manuscript should be suitable for publication. Most of my comments below and on the PDF version of the manuscript are meant to help the authors clarify some of their language.
- At a number of points in the manuscript, the authors simply to the “Geospatial Data Cloud” as the reference for some of their GIS model inputs. This is entirely unacceptable. The exact reference – both a URL and a data citation if available need to be included. Without this level of information, it is impossible to assess the quality of the data inputs into the model and hence the quality of the resulting output.
- The author need to better describe what is meant the term graded area ratio, it is used throughout the text and is an important component of many of the subsequent figures, but is somewhat unclear what it actually refers to or its overall importance in the computations. I believe it is simply what fraction of the study area is covered by that category (such as what fraction of the study area has southerly slopes) but I am not 100% sure my interpretation is correct.
- Need to indicate how surface roughness is computed including the algorithm (ideally with a citation).
- It is unclear what "more developed" means in this context. Does it mean that the vegetation tends to be higher or that there tends to be a different form/species of vegetation between northern and southern slopes. This is important in understand the susceptibility of different aspects to blowing snow.
- It is important to describe the DSSI in a bit better. The term prone is used without really defining what it means. I suspect it is in reference to the susceptibility of areas to snow accumulation due to wind drifting, but as this is the key index computed by the paper it is necessary to fully describe what you intend it to capture.
- It might be beneficial for the authors to describe the format of numerous figures that snow the Drifting and graded area ratios and weight of evidence lines as a function of the various properties in the models in a way that would guide their readers of how they should interpret the relationships presented in these figures.
Editorial suggestions
Page 1 Line 41 (and elsewhere) often winds is preferable to wind
Page 1 Line 43 (Need a month before 13, 2016)
Page 5 Figure 1. The figure requires an overview map to show the larger area of where the study area is located. The subunits should have their complete names and not abbreviations shown as most readers will not be familiar with them. The scale bar should probably be in kilometers for this journal. I do not think the title (Line Direction Map) ads much to the map itself.
Pages 5 & 6. The map on Figure 2 is impossible to view. It needs to be enlarged as do the dots indicating the locations of the monitoring sites. The authors should not simply rely on the autoplacement of the text but rather should manually place the descriptors of the stations so as not to obscure the train route.
Page 5 line 149 and elsewhere – the term “Geospatial Data Cloud” needs to be replaced with a reference to the exact URL or dataset on which the calculations are based. Without this it is impossible to assess the quality of the data inputs into the model.
Figure 13 is too small and hard to read. I would make each of its two panels identical in size and construct to the other figures that portray the ratios and weight of evidence calculations. This could be accomplished by stacking the two plots or separating them into two figures.
It might be best to rewrite the conclusions in paragraph form rather than numbered.
I have also made suggestion on the PDF of the manuscript itself as the page numbering is incorrect.
Author Response
Thank you very much for your valuable comments on the revision of the article. Based on your review comments, the specific revisions are as follows:
1,The factors involved in the spatial geospatial data cloud in the article are mainly environmental conditions, which are obtained from DEM data through GIS analysis. The query URL is as follows https://www.gscloud.cn/search
2,It may be that my expression is not accurate. The grading area ratio is actually a very simple concept, which is the ratio of the area of each subdivided secondary state to the total area of the secondary state. I have amended accordingly in the second paragraph of 4.2.
3,I have supplemented the surface roughness calculation method in Section 4.2.1 Part 3.
4, I have modified the slope aspect effect on vegetation and drifting snow in section 4 of 4.2.1.
5, I have restated DSSI in 4.3.1.
6, The meaning of the 3 curves in the figure has been explained in 2.1 and 4.2.
7, I have made modifications to Figures 1, 2, 13 and singular and plural.
8,I have modified the English style appropriately.
Reviewer 2 Report
The authors have done a case study of drifting snow susceptibility (DSS) based on the Weight of Evidence (WOE), backpropagation (BP), and genetic algorithm backpropagation (GA-BP) algorithms. Later these methods were fused to form WOE-BP and WOE-GA-BP models for the DSS. and GA.
The work is heavily GIS and ML-based modeling and falls under the journal scope. It is a very clever replication of susceptibility mapping work similar to landslide, forest fire, flood, groundwater, sinkhole, mineral, etc.:
• https://scholar.google.com/scholar?hl=en&as_sdt=0%2C5&q=susceptibility+Weight+of+Evidence&btnG=
• https://scholar.google.com/scholar?hl=en&as_sdt=0%2C5&q=susceptibility+back+propagation&btnG=&oq=susceptibility+back+propagra
• https://scholar.google.com/scholar?hl=en&as_sdt=0,5&q=susceptibility+genetic+algorithm+back+propagation
Given hundreds of replications of the WOE method in GIS, this paper tries to do something different aspect of susceptibility adopting the same old workflow, well known and redundant in the literature. It is project work for a small region with a small amount of data that only reports accuracy measures as a result and have local implication. It is really difficult to find the novelty in this study because there are a lot of similar previous studies with the same methodology and this research is just doing the same. The work used many machine learning models, training and validating the model, then applying it and evaluating the performance of each model. Finally, do the comparison and state the best model. This work will not have any contribution to the scientific community and should not be accepted in international journals, the authors should choose some regional journals to submit their paper.
It is unfortunate that early career researchers fall prey to easy susceptibility mapping publication rather than doing actual research with fieldwork and resolving problems in the real world with better implications. There are countless papers published over the last few decades since GIS became available looking at similar conditioning factors such as NDVI, Slope, aspect, profile, rainfall, etc. to map susceptibilities. They all read the same as this paper (abstract, body, method, and conclusion). They use methods A to Z to produce maps of similar accuracy and conclude one better than others. Certainly, the maps have value for the very local scale for which they have been created. Changing factors, data, and methods will have a very small impact on the accuracy, and implications are the same. If you see the following authors, all the contributions are the same with changing areas or methods:
• https://www.semanticscholar.org/author/B.-Pradhan/143620129
• https://www.semanticscholar.org/author/S.-Lee/39847097
• https://www.semanticscholar.org/author/H.-R.-Pourghasemi/2858616
• https://www.semanticscholar.org/author/D.-Bui/2382463
• https://www.semanticscholar.org/author/H.-Hong/27099181
• https://www.semanticscholar.org/author/B.-Pham/144109958
• https://www.semanticscholar.org/author/H.-Shahabi/48673769
• https://www.semanticscholar.org/author/B.-Ahmad/32082590
• https://www.semanticscholar.org/author/A.-Shirzadi/23155839
Papers from the above authors have been in the past decade. Certainly, the maps have value for the very local scale for which they have been created. However, the results are very specific to the given datasets and have only the conclusion of one better than another which does not give new information for the broader scientific community, and this does not warrant a publication.
How is this study different from than above-listed author’s work? Also, some authors are still replicating such works which are not good science, so do take them as references. Such a replication work should never be presented as an academic work unless it's a breakthrough application and has huge implications.
It is evident that graduate students need to have successful publications, but it should be some achievement to be proud of. This work is a good case study and has some implications in the area but cannot be an academic advancement in the field of geology. Again, these are technical skills to apply on a project site and should not be pursued as an academic thesis/publication by supervisors. Understanding soil, geology, and triggering thresholds with field and lab experiments are what research questions should answer that would be useful to those regions. Some references by authors on snow-related disasters and numerical modeling can be taken as a good example of research.
The authors’ use of WOE, BP, GA area well-known state statistical application with good probability estimation. After learning one skill set, it can be applied in some local projects or coupled with other aspects but don’t be like the above researchers accumulating replication works. Authors must know that factors can be changed, training data can be changed and there will always be a slight change in accuracy and that will not change any decision-making for resource management in the area. These works are very misleading to do real research and novel publication works. See the influence of those above authors, it’s always among each other. It is evident that academia needs to have successful publications, but it should be some achievement to be proud of, not replicating again and again.
1. The abstract needs improvement on the implication side. See journal template or https://mitcommlab.mit.edu/cee/commkit/abstract.
2. The authors’ use of ML and fusion techniques needs to be well justified with good literature backing their selection. Here only an AUC-based comparison has been presented. There are many such variations, each can be shuffled to compare accuracies https://rdrr.io/cran/caret/man/models.html.
3. Will this study change any decision-making for disaster management in the area? The recent flooding of such replication never considers such questions and this pattern has caused falsehoods to the new researchers in geology and mapping and has been misleading them to do real research and novel publication works. See the influence of those above authors contributing to each other.
4. The introduction is not built well, there is a minor literature gap on drifting snow hazards. But will mapping work along a small length of railway be good enough in the decision-making in the area? Does this will change in different data, methods, or areas are border validation needs to be examined. See https://mitcommlab.mit.edu/cee/commkit/journal-article/
5. It is unclear if there were any training and validation data set used. WOE and any ML method require training data to make the WoE values.
6. Selection of study area as an admin region is not a good choice and here it’s quite a small railway line. Snow events can cause from far away and how much such events have been observed to make such a case study area. The best practice is taking the wise region with the same elevation or higher such as watershed, then clipping out the final result for the line.
7. Train and test optimization is a must for BP and GABP.
8. Temporal resolution in the data selection process needs to be properly explained. If possible, add these to a table with all sources.
9. Also, varying classes of continuous fields in variables impact the result very much. So proper literature backing these intervals should be provided?
10. Discussion is missing. First snow and drifting patterns should be discussed, then susceptibility in detail, the reality in ground reality, natural factors, policy, development works, and activities comparing parallelly with the result. Please read properly what discussion should be in the journal’s template and https://mitcommlab.mit.edu/broad/commkit/journal-article-discussion/
11. Reporting accuracy and number from a computer model is output, not an outcome. Research work must have an outcome that has some information or implication.
12. Most citations are only done based on keywords and not their conclusion drawn or the basis that the research gap is built. Please understand the concept of the citation properly, use the one you read, understand, and conclude your statement written.
13. If possible show maps zoomed on the railroad sections.
14. Recent prediction works have also created black-box research which is not easily verifiable and available to readers. The data and model should also be provided for transparency and replication via a proper Zenodo or GitHub or official websites of lab/province/dept (e.g., https://northernchange.brown.edu/data-and-code/). It is highly appreciable that authors have added the data, but sharing of code/model for verification would be even great. FYI, https://authorservices.taylorandfrancis.com/data-sharing/share-your-data/data-availability-statements/
15. A minor ethical issue, authors in Asia add all the funding sources and collaborators in related/unrelated publications. As a general suggestion for the future, to improve the reliability and standard of research in Asia, unsolicited co-authorship and fundings should proceed with high integrity.is an imbalance in work and a missing author contribution. Please see the credit statement https://www.elsevier.com/authors/policies-and-guidelines/credit-author-statement for the proper and honest co-authorship.
Wish you all the best for future research work.
Author Response
Thank you very much for your comments!Please refer to the attachment for the specific revision of the article.

Reviewer 3 Report
This paper presents an evaluation of drifting snow susceptibility using GIS and a genetic algorithm back propagation (GA-BP) Algorithms. Authors used Xinjiang Afukuzhun Railway as a case study. They calculated the drifting snow susceptibility index through the weight of evidence (WOE) model, and used (GA-BP) algorithm to obtain an optimised evaluation index weights to improve the model's accuracy for the evaluation. Results show that the accuracy increases from the WOE model to WOE-BP and WOE-GA-BP models. The paper reads well but has room for the improvement. See the below:
- Mention the equation number.
- Elaboration the caption of figure 1.
- Same comment like comment-2 but for all figures.
- Section#2 needs to move.
- Few figures (for example Fig 17) are not up to the standard. It requires a major improvement.
- Mention the advantages and limitations of the model.
- Starting from 1 point didn't make much sense to start conclusions. Write few sentences before numbering the key findings.
Author Response
Thank you very much for your valuable comments on the revision of the article. Based on your review comments, the specific revisions are as follows:
1,I have numbered the formulas.
2, I have modified the title of Figure 1.
3, Although the description in Conclusion 2 is applicable to all numbers, the number of grids is the specific number selected in this study. Could you directly express it in this way?
4, I have reworked the typography.
5, Figure 17 is the correlation coefficient diagram of training samples, verification samples and total samples obtained after training by Matlab. I wonder which aspect of modification you refer to specifically?
6, The limitation description of the model is added in point 5 of Part 6.
7, A summary description was added at the beginning of Section 6, and point 1 was partially adjusted.
8,I have modified the English style appropriately.
Round 2
Reviewer 2 Report
The authors have replied to the comments with the changes made. However, the manuscript is kind of the same and does not feel any substantial improvement except for minor textual changes here and there. Although factors are fine, the resulting maps are highly doubtful as it is only susceptible towards the top. The basic concept that snowdrift susceptibility is important is because of how it occurs and how it causes damage. The damage in the western world esp in railway cases is due to deposition in tunnels, short circuits in switches and lines, and sometimes depositing along downward or leeward. Min wind speed is around 12mph for air to be able to lift snow particles, here high is fine, average does not play many roles, however, curvature could be something and wind direction and if there are any valley or pass along that high susceptible area. Also, here the tracks are fairly good hill plains and snowdrift will occur at that latitude and elevation but the deposition is not substantial and the damage across those plains tracks is almost null.
Given the effort of the authors, please take a step back and try to think the result, relate with the ground reality and does it make sense with this effort in the railway track of hilly plains.
One can present a small technical note or communication as this result is a very local case (very doubtful), cant be generalized (very specific) and has less impact (railway in plains). Again it is only input and out, no articulated research questions that are creative, address a research gap and have the potential to be impactful. See submission overview for technical note for RS in https://www.mdpi.com/journal/remotesensing/instructions and following for discussion are very important section to interpret the results. See following for conclusion and discussion:
- Discussion: Authors should discuss the results and how they can be interpreted from the perspective of previous studies and of the working hypotheses. The findings and their implications should be discussed in the broadest context possible and limitations of the work highlighted. It is also essential to discuss the strengths and limitations of one’s study. Comments on sources of uncertainty and error are appropriate for most papers. Future research directions may also be highlighted.
- Conclusions: This section is required. Conclusions should include precise, concise and quantitative statements about the significance of the study, highlight any new findings, and explain how the work could be extended in the future. Conclusions must be self-contained, a reader should not have to read through the paper to understand it and we recommend that acronyms are written in full. In the case of review papers, the conclusions should summarize the state-of-the-art of knowledge, knowledge gaps, and suggest ideas for future directions.
- Providing Author's contribution. https://www.elsevier.com/authors/policies-and-guidelines/credit-author-statement.
As a fully open access paper, the data and model should also be provided for transparency and replication else it will only lead to keyword-based citations but not the actual implementation of work. As a suggestion, please provide the data and model for transparency and replication via a proper Zenodo (https://zenodo.org/record/5899548#.Ye8mAXVKhhE) or Dryad Data (https://datadryad.org/stash/dataset/doi:10.5061%2Fdryad.12528) or GitHub (https://github.com/ronggz728/DNN_RNN_CNN_TPE) or official websites of lab/province/dept (e.g., https://northernchange.brown.edu/data-and-code/). Sharing code/model increases its implications and thus influence. FYI, https://authorservices.taylorandfrancis.com/data-sharing/share-your-data/data-availability-statements/.